# Mapping Amazon Forest Productivity by Fusing GEDI Lidar Waveforms with an Individual-Based Forest Model

**Luise Bauer** [1,*]**, Nikolai Knapp** [1,2] **and Rico Fischer** [1]

1 Department of Ecological Modelling, Helmholtz Centre for Environmental Research—UFZ Leipzig, Permoserstr. 15, 04318 Leipzig, Germany; nikolai.knapp@thuenen.de (N.K.); rico.fischer@ufz.de (R.F.)
2 Thünen Institute of Forest Ecosystems, Alfred-Möller-Str. 1, Haus 41/42, 16225 Eberswalde, Germany
* Correspondence: luise.bauer@ufz.de

**Abstract:** The Amazon rainforest plays an important role in the global carbon cycle. However, due to its structural complexity, current estimates of its carbon dynamics are very imprecise. The aim of this study was to determine the forest productivity and carbon balance of the Amazon, particularly considering the role of canopy height complexity. Recent satellite missions have measured canopy height variability in great detail over large areas. Forest models are able to transform these measurements into carbon dynamics. For this purpose, about 110 million lidar waveforms from NASA's GEDI mission (footprint diameters of ~25 m each) were analyzed over the entire Amazon ecoregion and then integrated into the forest model FORMIND. With this model–data fusion, we found that the total gross primary productivity (GPP) of the Amazon rainforest was 11.4 Pg C a$^{-1}$ (average: 21.1 Mg C ha$^{-1}$ a$^{-1}$) with lowest values in the Arc of Deforestation region. For old-growth forests, the GPP varied between 15 and 45 Mg C ha$^{-1}$ a$^{-1}$. At the same time, we found a correlation between the canopy height complexity and GPP of old-growth forests. Forest productivity was found to be higher (between 25 and 45 Mg C ha$^{-1}$ a$^{-1}$) when canopy height complexity was low and lower (10–25 Mg C ha$^{-1}$ a$^{-1}$) when canopy height complexity was high. Furthermore, the net ecosystem exchange (NEE) of the Amazon rainforest was determined. The total carbon balance of the Amazon ecoregion was found to be −0.1 Pg C a$^{-1}$, with the highest values in the Amazon Basin between both the Rio Negro and Solimões rivers. This model–data fusion reassessed the carbon uptake of the Amazon rainforest based on the latest canopy structure measurements provided by the GEDI mission in combination with a forest model and found a neutral carbon balance. This knowledge may be critical for the determination of global carbon emission limits to mitigate global warming.

**Keywords:** tropical rainforest; GEDI; lidar; forest model; FORMIND; carbon balance; structure; productivity

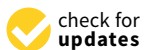



## 1. Introduction

Tropical rainforests represent an important carbon sink in the biosphere [1], so they have potential for mitigating global warming [2]. They account for about half of the carbon stored in global vegetation (350–600 Pg C) [1,3,4]. Unfortunately, estimates of carbon stocks and fluxes in the global carbon cycle are associated with large uncertainties [5]. For this reason, reducing uncertainties in the estimates of tropical carbon stocks and fluxes is one of the greatest challenges facing climate science. To better understand the global carbon cycle and thus global warming and climate change mitigation, it is necessary to conduct an accurate assessment of carbon fluxes in tropical forests, such as forest productivity [6]. However, forest productivity can be spatially heterogeneous and influenced by several forest attributes, such as canopy height, which can be a key variable [7,8]. Mapping canopy height complexity is therefore critical to understanding the history, function, and future of forest ecosystems [9].

The largest intact tropical forest is the Amazon rainforest [10]. In tropical America, for example, more than 80% of aboveground carbon is stored in forests [11]. However,

these forests are very heterogeneous and reveal complex structures. Thus, estimates for aboveground biomass in the Amazon rainforest diverge by a factor of two and range from 40 to 90 Pg C [12]. Based on the fact that variations in estimates of aboveground biomass in the Brazilian Amazon account for 60% of the variation in estimated net carbon flux for the region [13], it is important to estimate aboveground biomass and gross primary productivity for the Amazon as accurately as possible to determine the forests' carbon balance. Variations in aboveground biomass and productivity estimates primarily result from forest disturbance and recovery [14] but also from the limitations of the methods used so far.

Previously, aboveground biomass maps were mainly produced based on passive optical satellite data, such as Landsat or MODIS in combination with sparse lidar measurements from the Geoscience Laser Altimeter System (GLAS) located on the ICESat (Ice, Cloud, and land Elevation Satellite) [11,15–17]. However, there is a limitation to using measurements from passive optical satellites because they are only sensitive to tree cover, leaving the vertical structure of a forest unknown. Aboveground biomass estimates are thus saturated for high density forests. Lidar measurements, on the other hand, allow for the vertical structure of a forest to be measured.

A new generation of satellites could bring tropical forest surveying to a new level. GEDI is the first spaceborne lidar mission optimized for measuring ecosystem structures [15]. The lidar measurements of GEDI (with a diameter of ~25 m each) can be used for determining the forest canopy structure, forest aboveground biomass, and topography [15]. The dataset contains an average of 27 waveforms per 1 km$^2$. With millions of lidar waveforms in the Amazon, GEDI provides the opportunity to decrease inaccuracy in the determination of forest attributes such as aboveground biomass [18]. With the last spaceborne lidar sensor, ICESat GLAS, about 770,000 waveforms were available in the Amazon region [19], whereas in this study, about 110 million GEDI waveforms, all collected within 16 months, were compiled—exceeding the ICESat dataset by two orders of magnitude.

The aim of the study was to estimate the productivity and carbon balance of the entire Amazon by fusing GEDI measurements with forest simulations. Through the application of a dynamic forest model, it is possible to simulate the growth of a forest [20] and to extract the current state of a tropical forest by lidar waveform matching [19,21]. By merging the satellite data with the model simulations, the productivity and structure of the forest could be determined and analyzed. Uncertainties in productivity estimates could be reduced and studied in more detail by determining the structure, aboveground biomass, and growth of tropical forests. Therefore, the following questions are answered in this study:

- What are the total productivity and carbon balance of the Amazon rainforest and how are they spatially distributed?
- What are the relationships between forest productivity and other forest properties, such as aboveground biomass, canopy height complexity, net ecosystem exchange, and forest age in the Amazon rainforest?

## 2. Materials and Methods

### 2.1. Study Region

For this study, we considered the Amazon ecoregion in South America, with an area of $5.4 \times 10^6$ km$^2$. The spatial extent of the Amazon ecoregion was defined as all ecoregion polygons [22] of the "tropical and subtropical moist broadleaf forests" biome type, constrained by the Andes in the west, the Llanos and Atlantic Ocean in the north and east, and the Cerrado and dry forests in the east and south.

### 2.2. Lidar Data from the GEDI Mission

NASA's Global Ecosystem Dynamics Investigation (GEDI) is a large footprint lidar system that has been mounted on the International Space Station (ISS) since 2018 and generates waveforms. GEDI provides measurements of forest vertical structure in temperate

and tropical forests between 51.6° north and south latitude. The overall objective of the mission is to determine the effects of changing climate and land use on ecosystem structure and dynamics [15].

The GEDI laser system consists of three lasers: one that is split into two lasers (coverage laser) and two full-power lasers that remain unchanged. The tracks are located within a ~4.2 km wide strip spaced ~600 m apart. The footprint spacing per track is 60 m. The GEDI data consist of footprint and gridded datasets that contain information about the 3D properties of the vegetation. These data are associated with different levels of data processing. For the analyses in this paper, we used the Level2A data, which contain ground elevation, canopy top height, and relative height percentiles (here, we used a relative height at 95% (RH95)) [15].

The lidar data used in this study were collected worldwide between 2019 and 2020. Only the data located inside the Amazon rainforest ecoregion [22] were selected for this analysis. Each GEDI footprint has a diameter of ~25 m. The quality of each GEDI shot is checked by a "quality flag", which allows one to easily remove erroneous and/or lower quality waveforms [23]. After filtering, about 110 million individual GEDI measurements were available to study the structure of the Amazon rainforest. It should be noted that the temporal information in the GEDI data was not used in this study. The GEDI waveforms provide a structure-based snapshot of forest aboveground biomass and productivity that can be disentangled using the FORMIND forest model.

### 2.3. Individual-Based Forest Model FORMIND

FORMIND is an individual-based forest gap model that was developed in the late 1990s to simulate tropical forest dynamics in a composite of $20 \times 20$ m$^2$ patches ("gaps") and physiological processes at the tree level [20,24]. Therefore, the model is able to describe the growth development of tropical rainforests. Tree species are grouped into plant functional types. The main processes included in FORMIND are tree growth, tree mortality, recruitment, and tree competition. Trees mainly compete for light, expressed in the model as photosynthetic photon flux density. In addition, they compete for canopy space within their patches. With information about the successional state and environmental conditions, FORMIND enables upscaling to forest-wide carbon balances [25,26].

Here, FORMIND was applied to the Amazon rainforest, where species were assigned to three plant functional types (PFT) that differ in growth and mortality rate [12]. Then, FORMIND was used to estimate aboveground biomass (AGB), gross primary productivity (GPP), net ecosystem exchange (NEE), and age of trees. NEE is the difference between the ecosystem GPP and ecosystem respiration. The ecosystem GPP corresponds to the gross primary productivity of all trees in a forest. Ecosystem respiration consists of the autotrophic respiration of trees, as well as heterotrophic respiration from deadwood decay and soil organisms [26].

For the purpose of upscaling the forest model to the whole Amazon, the Amazon region was divided into environmental regions, and an extensive simulation was carried out for each region (see Section 2.4). Afterwards, the current forest state was determined from the simulations with the help of the GEDI waveforms and waveform matching (see Section 2.6). In this study, the established Amazon parameterization of FORMIND [12] was used.

### 2.4. FORMIND—Amazon Version and Regionalization

We assumed that forest dynamics are similar in areas with similar environmental factors [12]. The Amazon rainforest was therefore divided into regions with similar environmental conditions, as described in [12]. The considered environmental conditions were mean annual precipitation and mean annual photosynthetic photon flux density (PPFD) (available for the entire Amazon at a resolution of 0.5° (derived from WFDEI [27])), and clay content, available at a resolution of 8 km [28]. With these data, it was possible to assign each 1 km$^2$ sub-area in Amazonia to one of 1280 environmental regions, within each of



which homogeneous environmental conditions prevail [12]. For each environmental region, FORMIND was used to simulate 100 ha of forest dynamics over 1000 years, resulting in a total of 128,000 ha of simulated forest area. In the simulations, the drivers for mortality and photosynthesis were adjusted for each environmental region depending on the environmental conditions [12]. Additionally, simulated aboveground biomass dynamics are influenced by random stand-replacing disturbance events, e.g., fire, thus leading to a broad range of undisturbed and disturbed successional states [29]. From these simulations, forest parameters such as aboveground biomass (AGB), gross primary productivity (GPP), and net ecosystem exchange (NEE) were obtained.

*2.5. FORMIND Lidar Simulator*

FORMIND is also able to carry out virtual lidar campaigns for simulated forest stands. The lidar module integrated into FORMIND allows for the simulation of lidar point clouds and waveforms like the ones provided by GEDI [21]. In the lidar module, 3D space is represented by cuboidal voxels with a width of 0.5 m and a height of 1 m. To simulate the lidar waveform over a forest area, we created a tree list consisting of the generated data of a forest model and the positions and parameters of each tree. Then, a voxel representation of the forest was created so that voxels belonging to a tree and thus providing a contribution to the lidar waveform could be distinguished from empty voxels without contribution. The reflected energy of each voxel was modeled as an exponential decay function (Beer–Lambert's law) of leaf area above the voxel, and the horizontal energy distribution within the laser beam was modelled with a Gaussian function [30]. This lidar module was already tested for tropical forests via comparison with real lidar data [21].

Since the simulated lidar waveform had to be comparable to the GEDI waveform, the spatial scale of the GEDI lidar measurements was considered in the model. To do this, a circular footprint with a diameter of 25 m was cropped at the center of each simulated hectare in the model, reducing the simulated area from one hectare to a subplot. The energy of all voxels per 1 m height layer was summed up, and the derived waveform was finally normalized to the total sum to obtain the relative energy per height layer [19]. For each of the 1280 environmental regions, 100 waveforms (one per simulated hectare of forest) were simulated at 5-year intervals over a period of 1000 years, resulting in a total of 20,000 waveforms per environmental region.

*2.6. Waveform Matching*

Each real GEDI waveform was compared against hundreds of simulated waveforms to determine the state of the forest. This process is referred to as waveform matching [19]. Lidar matching was used to determine the current state of the forest under consideration of each GEDI waveform. Based on the forests simulated in FORMIND, virtual lidar waveforms were simulated and stored every five years. Each real GEDI waveform was compared to the dataset of all simulated lidar waveforms in the respective environmental region, and a relative overlap of the waveforms was determined starting at a height of 5 m [19,31]. The lowest 5 m waveforms were discarded in the waveform matching because the shapes of the real GEDI waveforms in these low heights are often influenced by uneven terrain and understory vegetation, leading to blurred ground return peaks. These influences were absent in simulated waveforms, which possessed sharp peaks at ground level.

The virtual waveforms with the best overlaps were used to determine the current forest condition, as these waveforms best matched the GEDI data and thus reflected reality (Figure S4).

Fifty simulated waveforms with the best overlaps were selected, with at least 70% overlap [19]. By linking the simulated data with the real data, it was possible to derive, for example, the simulation time steps of the best-matching waveforms and, from that, various forest parameters. For each of the 50 waveforms, the respective forest parameters were derived and then averaged over all 50 values (Figure 1). In this work, the focus was on the

productivity of the forest at a given time point, which was determined by the gross primary production. About 155 million GEDI waveforms in the considered area were available for analysis after the quality flag check. After rejecting some waveforms due to mismatches (no waveform matching possible because all overlaps <70%), and discarding waveforms whose RH95 values were <5 m, a total of about 110 million GEDI waveforms were included in our study. From these, it was possible to use waveform matching to determine, for example, the structure and productivity of the forest at the time the GEDI waveforms were recorded. Although most of our analyses were at the footprint level, for some results and figures, forest variables were aggregated to a 1 km$^2$ scale by averaging; 1 km$^2$ cell grids where no waveforms were located were not included in this analysis.

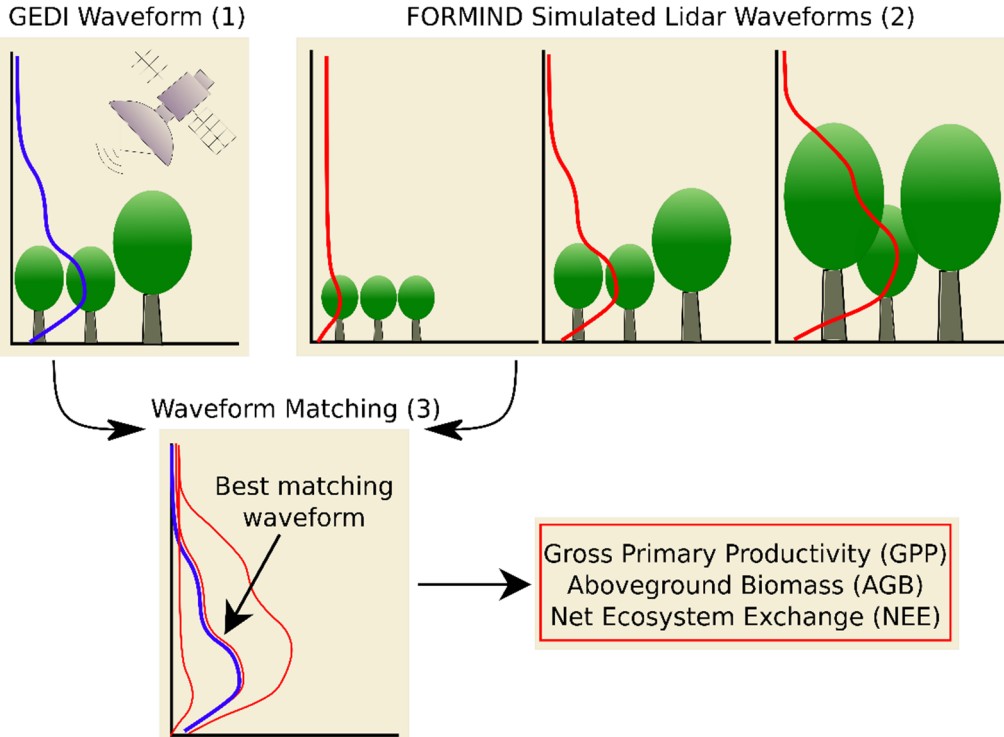

**Figure 1.** Workflow to combine the GEDI lidar data with the simulated lidar data from FORMIND. Each individual GEDI waveform (1) was considered and compared with lidar simulations from FORMIND (2). FORMIND samples lidar waveforms from over 1000 years of forest succession. The best 50 overlaps of the GEDI and simulation waveforms were determined (3). Each simulated waveform was associated with a simulated forest stand, which made it possible to infer various forest parameters from FORMIND. In this way, values for GPP, AGB, and NEE were calculated.

## 3. Results

### 3.1. Simulating the Potential Forest Dynamics for all Regions in the Amazon Rainforest

Forest simulations were generated for each environmental region in the Amazon. In total, a forest area of 128,000 hectare was simulated. As an example, the forest dynamics for two different regions, which mainly varied in precipitation and clay content, were compared (LPLC: low precipitation and low clay content; HPHC: high precipitation and high clay content) (Figure 2). In the HPHC region, the aboveground biomass ranged from 300–400 Mg ha$^{-1}$ of organic dry matter (odm) in the mature forest state (Figure 2a), with an average of 321 Mg odm ha$^{-1}$. The amount of AGB was reduced only by random stand-replacing disturbance events. In comparison, the LPLC region had a lower overall AGB (between 150 and 200 Mg odm ha$^{-1}$), where the average was 176 Mg odm ha$^{-1}$. In comparison to aboveground biomass, GPP and NEE differed much less for these two regions (Figure 2b,c). The mean values for GPP were 26 Mg C ha$^{-1}$ a$^{-1}$ for LPLC and

28 Mg C ha$^{-1}$ a$^{-1}$ for HPHC; the mean values for NEE $-0.2$ Mg C ha$^{-1}$ a$^{-1}$ for LPLC and 0.0 Mg C ha$^{-1}$ a$^{-1}$ for HPHC. The similarity in GPP for different regions could be attributed to different species (i.e., PFT) composition (Figure 2b). The LPLC case may have been dominated by very productive pioneer tree species, while the HPHC case may have been dominated by large shade-tolerant trees. Additionally, the NEE values were in the same order of magnitude for the two example regions, which can be explained by the fact that the simulated carbon balance of an old-growth forest was always balanced in this model framework as long as no further disturbances, climate change, or forest management took place (Figure 2c). However, when disturbances did occur, the dynamics of GPP and NEE were different for the two regions (see forest dynamics after peaks in Figure 2b,c).

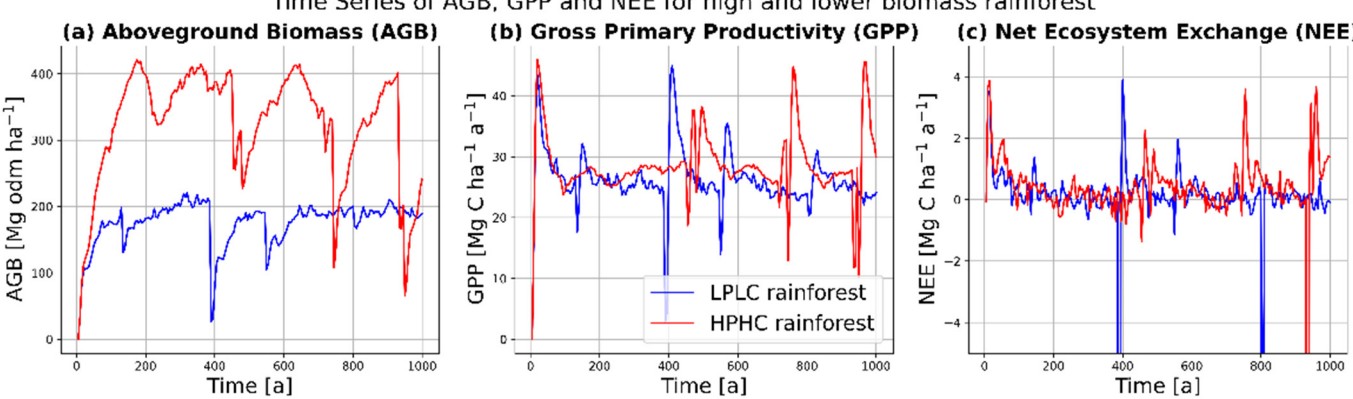

**Figure 2.** Simulated (**a**) aboveground biomass (AGB), (**b**) gross primary productivity (GPP), and (**c**) net ecosystem exchange (NEE) over simulation time for two exemplary environmental conditions (low precipitation, low clay (LPLC) and high precipitation, high clay (HPHC)). Blue: low precipitation (1750 mm a$^{-1}$), low clay content (15%), and medium PPFD (820 µmol m$^{-2}$ ha$^{-1}$). Red: high precipitation (3750 mm a$^{-1}$), high clay content (0.95), and high PPFD (980 µmol m$^{-2}$ ha$^{-1}$). For both simulations, random stand-replacing disturbance events were integrated.

### 3.2. Determining the State of the Forest with GEDI Lidar Data

More than 110 million lidar waveforms from NASA's GEDI mission were analyzed and linked with simulations from the forest model FORMIND. Information about the forest height was derived directly from each real GEDI waveform (here: RH95). Waveform matching between observed GEDI lidar data and forest simulations made it possible to estimate AGB, GPP, and NEE at the footprint level. The analysis was based on footprint level resolution. For visualization purposes only, the footprint-level resolution was aggregated to 1 km$^2$ and 20 × 20 km$^2$ (Figures 3 and 4). A frequency distribution of GEDI waveform counted over the 1 km$^2$ grid showed that the individual grid cells could contain between 1 and 100 GEDI waveforms (Figure S5).

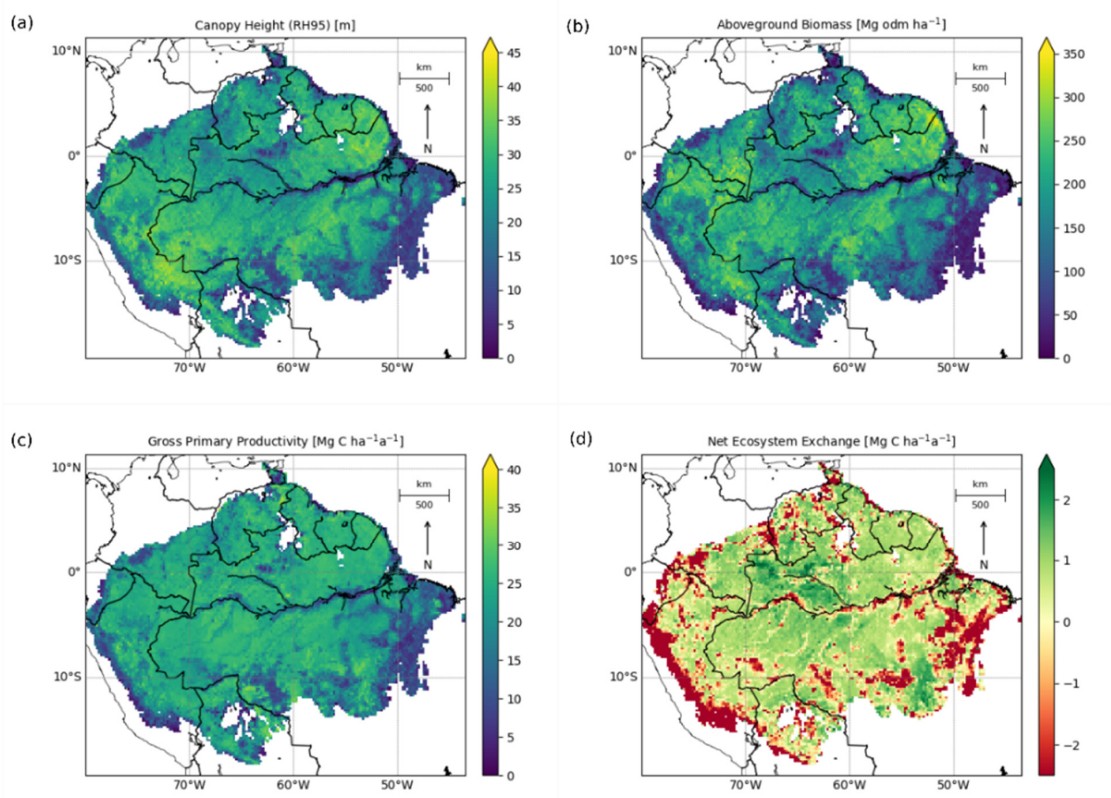

**Figure 3.** Maps of mean canopy height (RH95) in m (**a**), mean aboveground biomass (AGB) in Mg odm per ha (**b**), mean gross primary productivity (GPP) in Mg C per ha per year (**c**), and net ecosystem exchange (NEE) in Mg C per ha per year (**d**). The 1 km$^2$ resolution was aggregated a to 20 × 20 km$^2$ resolution for visualization purposes.

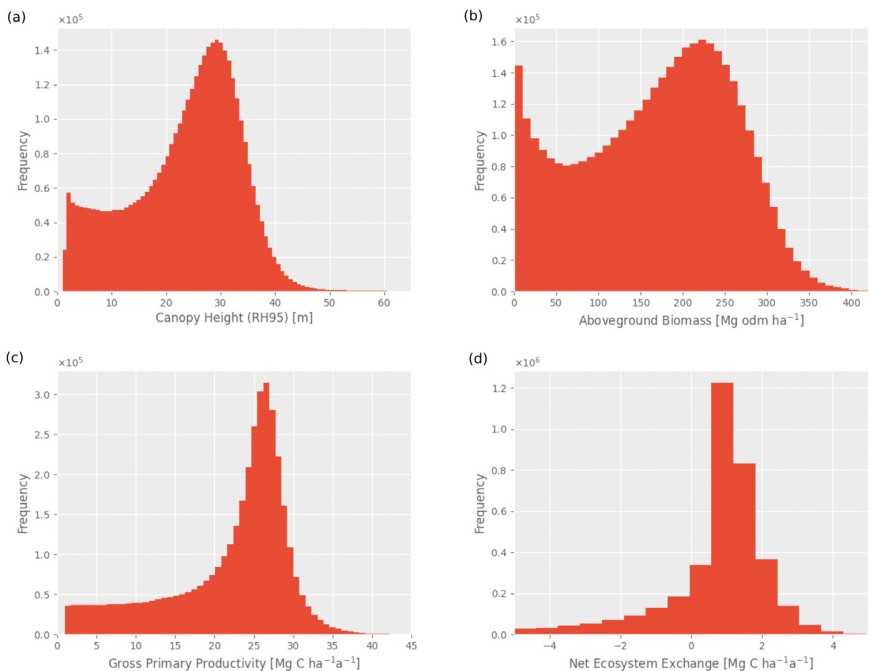

**Figure 4.** Frequency distributions of canopy height (RH95) in m (**a**), mean aboveground biomass (AGB) in Mg odm per ha (**b**), mean gross primary productivity (GPP) in Mg C per ha per year (**c**), and net ecosystem exchange (NEE) in Mg C per ha per year (**d**) associated with the maps in Figure 3, with a resolution of 1 km$^2$ for the Amazon based on 110 million lidar waveforms.

In our results we obtained forest heights that ranged from 0 m to a maximum of 50 m in height (Figures 3a and 4a). The mean forest height in the Amazon was $23.4 \pm 10.2$ m at a spatial resolution of 1 km$^2$. After applying the data–model fusion, the estimated aboveground biomass had a mean value of $163.4 \pm 93.1$ Mg odm ha$^{-1}$ and could reach values of up to 400 Mg odm ha$^{-1}$ (Figures 3b and 4b). These high aboveground biomass areas were located in the northeast, the Guiana Shield, and the central Amazon. Low aboveground biomass values were most evident in the southeast, where the Arc of Deforestation is located. Spatial variations could also be observed in the GPP distribution of the Amazon (Figures 3c and 4c). Here, values ranged from 0 to 40 Mg C ha$^{-1}$ a$^{-1}$, with a mean value of $21.1 \pm 8.7$ Mg C ha$^{-1}$ a$^{-1}$. GPP was low in forested areas where aboveground biomass was also low. The productive forest areas were mainly in the central Amazon Basin. The previously described distributions of high and low values of aboveground biomass and GPP, respectively, could also be seen in the NEE distribution (Figures 3d and 4d). Negative NEE values represented carbon sources, i.e., where carbon emissions due to respiration could not be compensated for by forest carbon uptake (please note: direct carbon emissions due to deforestation were not considered). The NEE map of the Amazon showed that large areas are carbon sinks. Only regions—such as in the Andes in the west and the Negro–Branco moist forests in the north, and the southeast—where the Arc of Deforestation is located were identified as carbon sources. There, up to 2.5 Mg C ha$^{-1}$ carbon was found to be emitted annually. In the Amazon Basin, on the other hand, there are carbon sinks that absorb up to 2.5 Mg C ha$^{-1}$ per year. Overall, we found that the Amazon rainforest has a neutral carbon balance, with a total NEE value of $-0.1$ Pg C a$^{-1}$. In our calculations, no direct carbon emissions from deforestation were considered (carbon emissions from deforestation: approximately $-0.46$ Pg C a$^{-1}$; see [11]), but the degradation of forests and resulting carbon emissions were already included.

The estimated carbon turnover time $\left( = \frac{AGB}{NPP} \right)$ for our analysis was 42 years. The turnover time varied from 0 to 120 years. Especially in the northeast and along the border between Peru and Brazil, long turnover times of 70–80 years could be observed, while in the Rio Negro Campinarana area and the southeast, near the Arc of Deforestation, short turnover times of 15–20 years occurred (see Figures S1 and S2).

Means and standard deviations for the different forest parameters, as well as Amazon-wide totals for the forest parameters, were calculated. The mean values were calculated from all waveforms located in a 1 km$^2$ cell, and the standard deviations were calculated accordingly. As an overview, the results are shown in Table 1.

**Table 1.** Mean values and standard deviations at a lidar footprint resolution of 25 m diameter, mean values and standard deviations at a map resolution of 1 km$^2$, and Amazon-wide totals (5.4 Mio km$^2$) for RH95, AGB, GPP, NPP, turnover time, respiration, and NEE. Here, turnover time is defined as AGB/NPP, and respiration is defined as GPP minus NPP (autotrophic respiration).

| | Mean ± Standard Deviation at Lidar Footprint Resolution (25 m Diameter) | Mean ± Standard Deviation at Map Resolution (1 km$^2$) | Total Amazon (5.4 Mio km$^2$) |
|---|---|---|---|
| Number of GEDI waveforms per Area | 1 | $27 \pm 17$ | 109,696,420 |
| RH95 | $23.7 \pm 12.4$ m | $23.4 \pm 10.2$ m | 23.4 m |
| AGB | $175.5 \pm 118.6$ Mg odm ha$^{-1}$ | $163.4 \pm 93.1$ Mg odm ha$^{-1}$ | 88.3 Pg odm |
| GPP | $21.5 \pm 11.2$ Mg C ha$^{-1}$ a$^{-1}$ | $21.1 \pm 8.7$ Mg C ha$^{-1}$ a$^{-1}$ | 11.4 Pg C a$^{-1}$ |
| NPP | $4.0 \pm 2.1$ Mg C ha$^{-1}$ a$^{-1}$ | $3.9 \pm 1.4$ Mg C ha$^{-1}$ a$^{-1}$ | 2.1 Pg C a$^{-1}$ |
| Turnover Time | $43.9 \pm 56.5$ a | $42.1 \pm 64.5$ a | 42.1 a |
| Respiration | $17.5 \pm 9.1$ Mg C ha$^{-1}$ a$^{-1}$ | $17.2 \pm 7.5$ Mg C ha$^{-1}$ a$^{-1}$ | 9.3 Pg C a$^{-1}$ |
| NEE | $-1.4 \pm 13.0$ Mg C ha$^{-1}$ a$^{-1}$ | $-0.3 \pm 5.3$ Mg C ha$^{-1}$ a$^{-1}$ | $-0.1$ Pg C a$^{-1}$ |

### 3.3. The Role of Canopy Height Complexity for the Aboveground Biomass and Productivity

The relationship between aboveground biomass (AGB), productivity (GPP), and the heterogeneity of canopy height complexity was investigated (Figure 5a). Here, canopy height complexity was described by the standard deviation of the RH95 of all GEDI lidar shots within 1 $km^2$. Hence, forests with a rather homogeneous canopy height had a low complexity, and forests with a heterogeneous canopy height had a high complexity. First, it could be seen that there was no clear trend in the data (Figure 5a). Only when looking at canopy height complexity was it noticeable that GPP increased from 0 to 45 Mg C $ha^{-1}$ $a^{-1}$ up to an AGB of 150 Mg odm $ha^{-1}$ when canopy height complexity was low. In forests with an aboveground biomass of larger than 150 Mg odm $ha^{-1}$ and a low canopy height complexity, productivity decreased again. Heterogeneous forest canopy heights, on the other hand, showed a positive trend and had productivity between 5 and 20 Mg C $ha^{-1}$ $a^{-1}$ for aboveground biomasses between 100 and 300 Mg odm $ha^{-1}$.

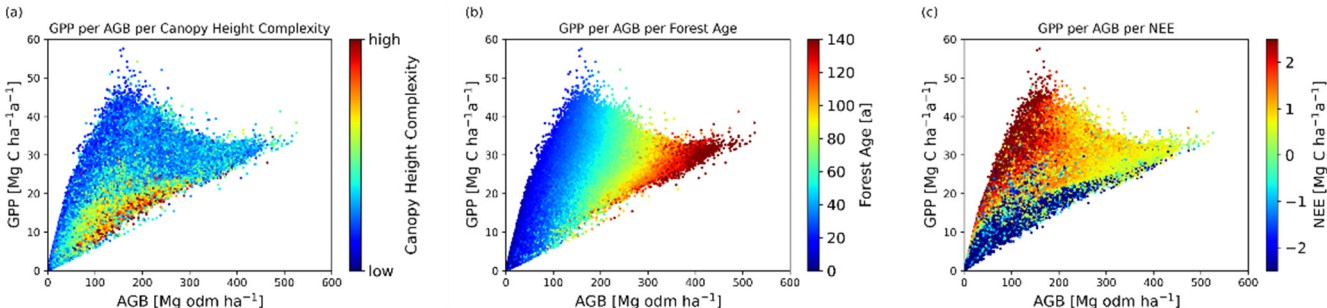

**Figure 5.** Relation between forest aboveground biomass (AGB) in Mg C $ha^{-1}$ and productivity (GPP) in Mg C $ha^{-1}$ $a^{-1}$. One point corresponds to a forest stand with an area of 1 $km^2$, whereby only areas containing more than 20 GEDI waveforms were considered. The points are colored (**a**) according to the heterogeneity of canopy height (SD of RH95) within this 1 $km^2$, (**b**) according to the mean age of the simulated forest within this 1 $km^2$, and (**c**) according to the net ecosystem exchange (NEE) within this 1 $km^2$, where positive values show a carbon sink and negative values show a carbon source. Mean forest age was calculated as the basal area weighted average of all tree ages.

The behavior could be described by different age structures (Figure 5b) of forests. Young forests (<40 years) with aboveground biomass up to 150 Mg odm $ha^{-1}$ showed high productivity and older forests (>40 years) with aboveground biomass above 150 Mg odm $ha^{-1}$ had reduced productivity. Additionally, considering the relationship between GPP, AGB, and NEE (Figure 5c), it can be noted that forests with the lowest productivities along the whole aboveground biomass range showed negative NEE values (carbon source behavior). Among the more productive forests, the highest positive NEE values (strongest carbon sinks) were observed for forests with low aboveground biomass (<150 Mg odm $ha^{-1}$), while forests with high biomass (>150 Mg odm $ha^{-1}$) tended to have neutral carbon balances.

It can be concluded that, for the same aboveground biomass, early-successional forest stands have higher GPP values, stands with low canopy height complexities have higher GPP values, and stands with positive NEE also have higher GPP values. This suggests that a lower structural complexity indicates early-successional forest stands with carbon sink behavior.

In addition, the relationship between AGB, GPP, and canopy height complexity was considered in terms of precipitation (Figure S3). For this purpose, the data were divided into low, medium, and high precipitation and compared. It was noticeable that in areas of low precipitation (Figure S3a), the aboveground biomass reached a maximum of 350 Mg odm $ha^{-1}$, while in medium (Figure S3b) and high (Figure S3c) precipitation regions, the aboveground biomass reached 500 Mg odm $ha^{-1}$ in some parts. Furthermore, the productivity of forests in medium precipitation areas was significantly lower. The maximum values here were around 40 Mg C $ha^{-1}$ $a^{-1}$; in comparison, productivity

reached a maximum of 45–50 Mg C ha$^{-1}$ a$^{-1}$ in low precipitation areas and a maximum of 50–60 Mg C ha$^{-1}$ a$^{-1}$ in high precipitation areas.

### *3.4. Comparison of Our Result with Previous Studies*

We compared the results of our study with those of previous studies for aboveground biomass, gross primary productivity, and net ecosystem exchange. Data from Rödig et al. [12], Rödig et al. [25], Malhi et al. [32], the MODIS GPP satellite, and the eddy flux tower [33,34] were used for comparison.

#### 3.4.1. Aboveground Biomass

For the aboveground biomass comparison, we used the biomass map from Rödig et al. [12] and compared the frequency distributions of the two datasets (both datasets being cropped to the same spatial extent; Figure 6).

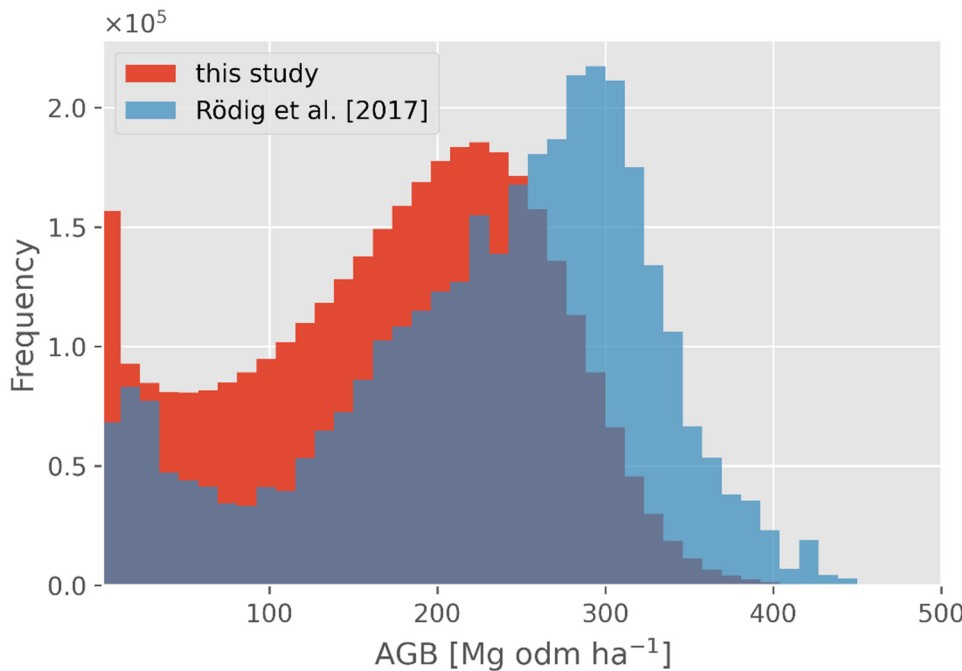

**Figure 6.** Frequency distribution of aboveground biomass of this study, consisting of the combination of GEDI and FORMIND data and the aboveground biomass map from Rödig et al. [12], with a resolution of 1 km$^2$ for the Amazon Basin.

In this study, more aboveground biomass values were observed in the range between 0 and 200 Mg odm ha$^{-1}$ compared to [12]. The peaks of the frequency distributions were around 250 Mg odm ha$^{-1}$ for this study and 300 Mg odm ha$^{-1}$ for Rödig et al. [12]. A comparison on the Amazon area can be seen in Figure S6. High differences could be observed in the border area of the central Amazon Basin, whereas the data within the central Amazon Basin matched well.

#### 3.4.2. Gross Primary Productivity

For a comparison of GPP, the GPP map of Rödig et al. [25], field data from Malhi et al. [32], and the GPP product from MODIS (MOD17A2HGF, which are gap-filled 8 day sums of GPP acquired by the Terra satellite; here aggregated to 1 km$^2$ from annual averages for the period from the beginning of 2019 to the end of 2020 [35]) were analyzed (Figure 7). For all three studies, the GPP values between 20 and 30 occurred most frequently. However, the distribution from this study was closer to the MODIS-based distribution than the Rödig-based distribution. The values from the study of Rödig et al. [25] were mostly between 22 and 27 Mg C ha$^{-1}$ a$^{-1}$, those from MODIS ranged between 25 and 32 Mg C ha$^{-1}$ a$^{-1}$,

and those from this study ranged between 22 and 30 Mg C ha$^{-1}$ a$^{-1}$. However, a wider range of GPP values was observed in this study (0–40 Mg C ha$^{-1}$ a$^{-1}$) in comparison to MODIS product (values between 15 and 35). The data of Malhi et al. [32] showed higher values than the average GPP determined in this study, but this was reasonable because their amount of included forest inventory data was rather low, the forest plots were much smaller in size than the resolution of the our map (1 km$^2$), and the locations of these forest plots were often in productive forests and not randomly selected [36].

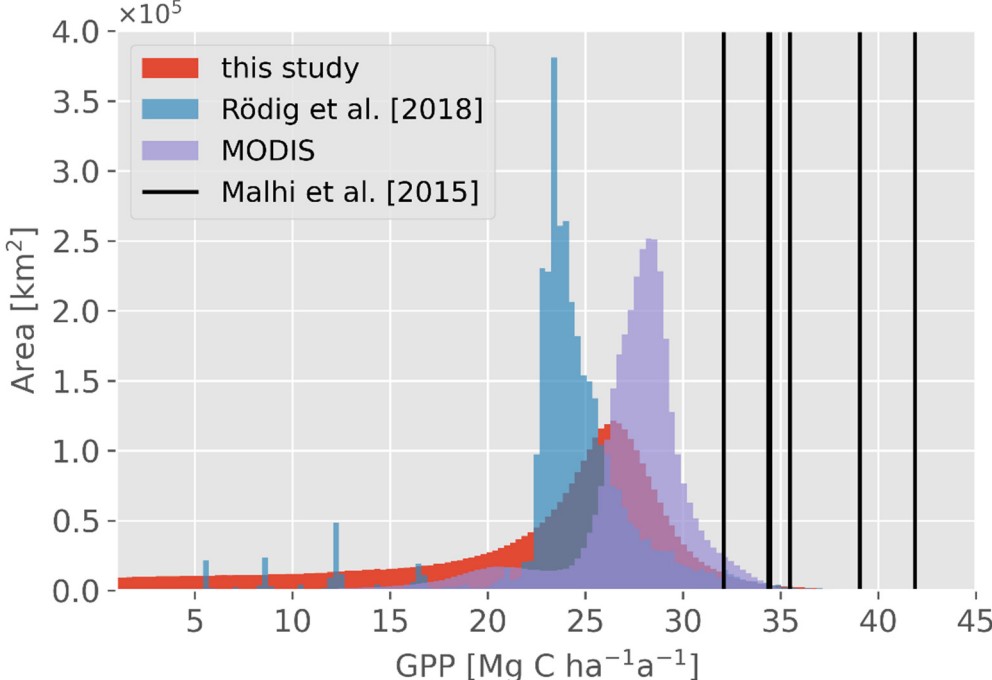

**Figure 7.** Absolute frequency distribution of GPP values determined in this study compared to the results from Rödig et al. [25], the MODIS-derived GPP product, and forest inventory data from Malhi et al. [32] for the Amazon Basin. All maps have a spatial resolution of 1 km$^2$ or were aggregated to 1 km$^2$ by averaging (MODIS). In addition, the same area of interest was considered for all maps (Amazon; 5.4 million km$^2$).

For a pixel-to-pixel comparison of GPP, the GPP map of this study was compared to those of Rödig et al. [25] and MODIS in the Amazon Basin (Figure S7a,b). It can be seen from both comparisons that the differences in the central Amazon Basin were rather small ($\pm 5$ Mg C ha$^{-1}$ a$^{-1}$). Only in the southern part and the Arc of Deforestation can larger deviations be observed ($\pm 10$ Mg C ha$^{-1}$ a$^{-1}$).

### 3.4.3. Net Ecosystem Exchange

For a comparison of our derived NEE map, we used aggregated values from eddy covariance flux measurements from GF-Guy [33] and BR-Sa3 [34] (Figure 8). Both measurements were within the range of the identified data of this study.

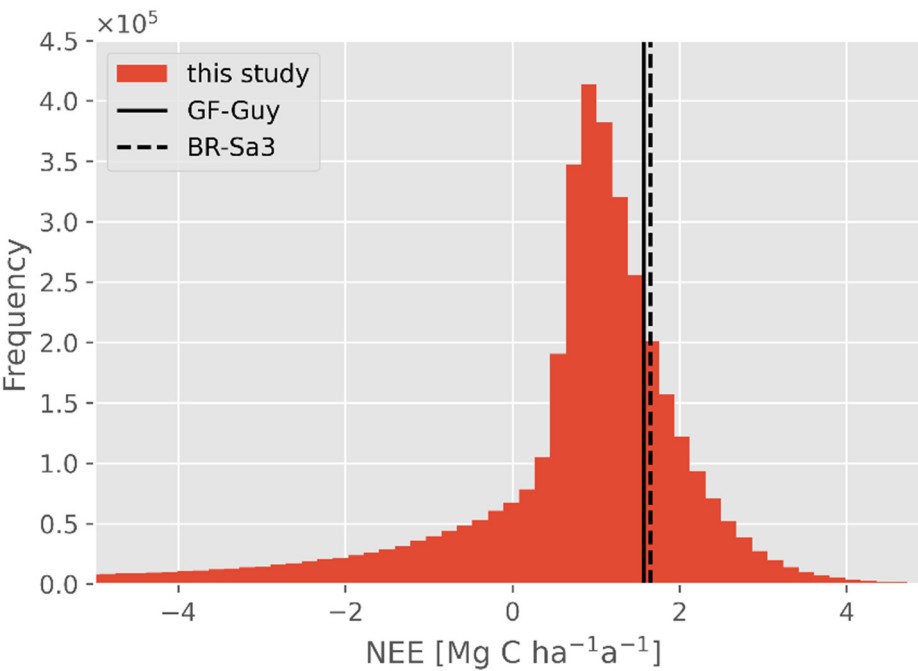

**Figure 8.** Frequency distribution of NEE, determined in this study by combining GEDI and FOR-MIND data, compared to eddy flux data from GF-Guy and BR-Sa3. The data of this study are available in 1 km$^2$ resolution. Eddy flux data were averaged values from Bonal et al. [33] and Goulden et al. [34], and they are plotted as values (GF-Guy: 1.57 Mg C ha$^{-1}$ a$^{-1}$; BR-Sa3: 1.65 Mg C ha$^{-1}$ a$^{-1}$).

## 4. Discussion

### 4.1. The Carbon Dynamics of the Amazon

Here, it was demonstrated that an Amazon-wide determination of gross primary productivity is possible with linkage from GEDI lidar data and simulations from the individual-based FORMIND forest model. By fusing the data, we obtained a variety of forest parameters that were examined in terms of canopy height complexity. In this study, two main results were obtained. First, the structure-related relationship between aboveground biomass, net ecosystem exchange, forest productivity, and forest age was explored. Second, by linking the simulation data with the satellite data, an Amazon-wide carbon balance map was created. This map shows, among other things, that certain areas in the Amazon have already turned into a carbon source, while other areas are carbon sinks. Overall, according to this analysis, the Amazon can be considered to have a neutral carbon balance with a total NEE of $-0.1$ Pg C a$^{-1}$. This NEE value is close to carbon neutrality.

It should be noted that our net ecosystem exchange did not include direct carbon emissions from disturbances such as deforestation. The authors of [37] established a carbon balance in which South America, among other regions, was considered. In the carbon balance, the net carbon exchange was split into individual components. Emissions from fires in South America were estimated using two datasets—Global Fire Emissions Database Version 4.1 (GFED4) and Global Fire Assimilation System (GFAS)—that considered the area burned and the radiative power of fires, respectively. For example, emissions from fires in South America were 0.325 Pg C a$^{-1}$ [37]. If these emissions were included in our NEE calculations, the Amazon would be considered to be a carbon source. If deforestation and degradation continue in the future, the Amazon risks becoming a larger carbon source.

Predictions that the Amazon may become a carbon source or that the decline in aboveground biomass will become irreversible are already supported by other publications. The author of [38] investigated the spatio–temporal dynamics of forest carbon in the Brazilian Amazon in the period of 2010–2019 using satellite data. Qin et al. [38] observed that the changes in forest area in the Amazon have different causes, such as the extreme El Nino year in 2015 and drought in 2019 [38]. Aboveground biomass declines in the Brazilian

Amazon are caused by deforestation, forest fragmentation, forest fires, and mortality from climatic disturbances such as storms and drought [38]. Even though AGB losses in areas with intact forests were only 0.10 Mg C ha$^{-1}$ a$^{-1}$ in 2010–2019, Qin et al. [38] drew attention to the fact that ongoing land-use changes, increasing climate extremes in the coming decades, and new Brazilian government policies may reduce the ability of forests to absorb carbon. They warned that achieving the goals of the REDD+ program will become more difficult.

Although the Amazon is generally still considered as a carbon sink, there have been few studies that predicted that the Amazon is a carbon source [39]. Our NEE map (Figure 3d) showed many areas where the Amazon is already a carbon source. A similar behavior with a similar distribution could be seen in [40], where it was found that large parts of the Amazon are already so disturbed that they are sources of carbon; these are mainly the areas of the Arc of Deforestation in the southeast of the Amazon. Gatti et al. [41] also studied the Amazon's carbon footprint and saw the risk of the Amazon becoming a source of carbon. In 2011, they noted that the Amazon Basin was still a net carbon sink of (0.25 ± 0.14) Pg C a$^{-1}$, but they now suspect a trend that the Amazon may become a carbon source due to emissions from fires and drought [41].

### 4.2. Relationship between Forest Dynamics and Structure

In this study, it was found that the individual-based forest model in combination with the GEDI data was well suited to investigate the vertical and horizontal structure of the forest. Thereby, hidden relationships between dynamics and structure became evident. As Rödig et al. [12] already mentioned, a combination of remote sensing data with forest models provides a better understanding of forest growth and the impact of canopy height complexity [12].

Here, it was found that forests with medium aboveground biomass and a heterogeneous horizontal structure have lower productivity than forests with a homogeneous horizontal structure. Heterogeneous canopy height structure probably results from a mixture of old and young forest patches (or disturbed and undisturbed), consisting of large trees shading smaller trees and some young high-productivity trees. Homogeneous canopy height structures are composed of many medium-aged trees, which are associated with high GPP [42].

The extensive dataset of the GEDI mission made it possible to generate a detailed forest height map. Due to the high resolution of the lidar data, spatial patterns could be recognized. Between both the Rio Negro and Solimões rivers, as well as in the transitional forests from the southern part of the Amazon to the Arc of Deforestation, medium tree heights from 20 to 35 m could be seen. Likewise, the significantly higher tree areas in the northeast and western Amazon were clearly visible with tree heights from 30 to 45 m. These results were also noted in the research of Saatchi et al. [43]. In [43], the structure of the forest was described and studied using canopy height. The canopy height (RH90) from the ICESat mission was used in [43] to show the distribution of tall trees, potential gradients, and large spatial variability. The results are therefore comparable, although the extensive dataset of the GEDI mission made it possible to resolve significantly more lidar waveforms per km$^2$, which could lead to more accurate results.

Mitchard et al. [44] used ground plots to calculate the aboveground biomass and, consequently, total carbon stock in the Amazon. They compared results from their ground plot with remote sensing results generated in [11,43]. Mitchard et al. [44] criticized the mismatches between ground plots and remote sensing maps. There are severe under- or overestimations of aboveground biomass. They [44] argued that interpretation from remote sensing data using single relationships between tree canopy height and aboveground biomass led to large, spatially correlated errors. This challenge was addressed in this study, and we determined the forest parameters, such as the AGB, GPP, and NEE, of the Amazon by linking over 110 million GEDI waveforms with the forest model FORMIND. For this, we did not just use a single metric describing the tree canopy height; instead we used the

complete GEDI waveform at each GEDI shot position to determine the forest state. In our study, the entire information from the waveform was used, which substantially improved the description of the current forest state and thus reduced ambiguities (see, e.g., [19]).

For future investigations, in order to follow up on this analysis and to examine the Amazon-wide forest structure and parameters in more detail, it makes sense to also analyze the horizontal structure of the forest at a high resolution. There are already radar missions that can detect the horizontal forest structure from wall to wall (e.g., TanDEM-X and BIOMASS) [45,46]. By linking GEDI data, radar data, and forest models, it is possible to conduct a detailed investigation of forest structure in order to explore new, yet unknown forest parameter connections.

## 5. Conclusions

This study answered two main questions about the total productivity and carbon balance of the Amazon rainforest (and their spatial distribution) and about the relationship between forest productivity (GPP) and various forest attributes, such as aboveground biomass, net ecosystem exchange, canopy structure, and age.

Low canopy height complexity was linked to young to middle-aged forest stands with high GPP values and neutral-to-positive carbon balance. It is important to protect these forests from deforestation and degradation in order to continue storing carbon so that climate change can be partially mitigated.

In addition, our analysis showed that the total productivity and total carbon balance of the Amazon rainforest were 11.4 and $-0.1$ Pg C a$^{-1}$, respectively. Low gross primary productivity was found in the southeast, where the Arc of Deforestation is located, and high gross primary productivity was found in the central Amazon Basin. A negative net ecosystem exchange was mainly found in the Andes, the Negro–Branco moist forests, and the Arc of Deforestation. A positive NEE was mainly found in the central Amazon Basin. Furthermore, Amazon-wide maps were created, showing the forest parameters forest height, aboveground biomass, gross primary productivity, and net ecosystem exchange. They directly showed the regions in the Amazon that have been disturbed due to various environmental and anthropogenic influences such as fires, deforestation, or drought and therefore have provided little aboveground biomass, gross primary productivity, or net ecosystem exchange. Large parts of the central Amazon Basin appear still intact. However, even there, the first signs of degradation are already visible. In our analysis, the Amazon still had a neutral carbon balance. To convert it back into a carbon sink, it is necessary to support climate change mitigation. To ensure that the Amazon remains the largest intact rainforest on Earth, deforestation and degradation must decline. High spatial resolution knowledge of Amazon rainforest behavior as a carbon sink or source can help policymakers make localized decisions for climate change mitigation actions and forest restoration programs.

**Supplementary Materials:** The following are available online at https://www.mdpi.com/article/10.3390/rs13224540/s1, Figure S1: Map of turnover time, Figure S2: Frequency distribution of turnover time for the Amazon based on 110 million lidar waveforms, Figure S3: Comparison of aboveground biomass with gross primary productivity for three different precipitation areas, Figure S4: Representation of the waveform matching, Figure S5: Frequency distribution for the number of GEDI waveforms per 1 km$^2$ grid cell, Figure S6: Map of the difference between biomass from this study and data from Rödig et al. [12] for a resolution of 1 km$^2$, Figure S7: Map of the difference between GPP from this study and data from Rödig et al. [25] and MODSI data for a resolution of 1 km$^2$.

**Author Contributions:** Conceptualization, L.B. and R.F.; methodology, L.B., R.F. and N.K.; software, L.B. and N.K.; validation, L.B.; formal analysis, L.B.; investigation, L.B.; resources, L.B., N.K. and R.F; data curation, N.K. and L.B.; writing—original draft preparation, L.B; writing—review and editing, L.B., R.F. and N.K.; visualization, L.B.; supervision, R.F.; project administration, R.F.; funding acquisition, R.F. All authors have read and agreed to the published version of the manuscript.

**Funding:** This study was financially supported by the German Federal Ministry for Economic Affairs and Energy (BMWi) under the funding code 50EE1911.

**Data Availability Statement:** Publicly available datasets were analyzed in this study. These data can be found here: https://e4ftl01.cr.usgs.gov/GEDI/, accessed on 29 March 2021. The simulation data were generated with FORMIND. The forest model can be downloaded from https://formind.org/model/, accessed on 16 March 2021.

**Acknowledgments:** We would like to thank the reviewers for their thoughtful comments and efforts towards improving the manuscript. This study was financially supported by the German Federal Ministry for Economic Affairs and Energy (BMWi) under the funding code 50EE1911.

**Conflicts of Interest:** The authors declare no conflict of interest.

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
