# Peer review of "Mapping Amazon Forest Productivity by Fusing GEDI Lidar Waveforms with an Individual-Based Forest Model"

_remotesensing, doi:10.3390/rs13224540_

Round 1

Reviewer 1 Report

The authors present a very interesting paper on estimating AGB and productivity in the Amazon rain forest using GEDI Lidar data and a forest model.

Except for two minor point, the manuscript can be accepted in present form.

  • In the first sentence of the abstract, you state that the "Amazon rainforest is considered as a carbon sink", but as you say in the discussion, this view is now contentious, so you should weaken this statement somewhat.
  • In line 211 you use the abbreviation "odm" for the first time, please explain it.

Reviewer 2 Report

This study retrieves forest state (AGB, GPP and NEE) based on forest vertical structure by combining an individual-based forest succession model, lidar simulation and GEDI waveforms. The work is interesting and meaningful. But the presentation, interpretation and discussion of the results need quit a bit of improvements.

First,  the conclusion does not speak to the results and discussion... The introduction claims two major questions ARE answered by this study. But the conclusion does not seem to give the answers based on your results. All very generic. Futhermore, Line 407-409. I don't even see any support from your results and discussion about this conclusion point.... how did you reach this conclusion? Higher productivity (GPP) surely does not mean higher carbon sequestration. In fact, you results and discussion do not speak much about the NEE, but mostly AGB and GPP.

Second, in the abstract, the “forest structure” is unclear. the statement on line 20-21 seems to suggest we need homoegenous forest stands for higher productivy,  which is misleading. Pleasse be specific about what you mean by “forest structure” here. It is heterogeneity or complexity of canopy heights. Actually I’d choose to use canopy heights rather than structure in general whenever you speak to your results since your results are specifically about the complexity of canopy heights.

Third, why NEE is not presented in the results an discussion in the same way as GPP in relatioin to AGB? It looks like to me that you actually wanted to speak more about NEE, judging from your claim on line 407-409. If possible, either add more analyses about NEE, e.g., a figure like fig. 5 but for NEE. Or simply focus on AGB and GPP. Rather than making conclusions about NEE that are not substantiated by your own results in this study.

Fourth, validation.  I understand this is a perpetual problem for model-based studies. However, some attempt, albeit flawed, is still warranted. The ref. [19] presented some validation about this approach. But I see that some validation possibility by comparing the distribution of model-based canopy height with the distribution of GEDI-based height in the 1km2 grids. If no validation at all, you need to be explicit about the potential flaws in this model-based estimates. I see no section dedicated to this issue on validation.

Detailed comments,

  1. Line 113-114, What does this claim mean? GEDI waveforms have no info. about when they are acquired?
  2. Line 155, interpolation of what data at what resolution into what resolution using what interpolation method?
  3. Line 155-156, Is each 1km2 or 100 ha randomly located in each of the 1280 environmental regions?
  4. Line 156-158, Is FORMIND parameterized differently for each 100-ha region? Good to be explicit here than just referring to [12] since this seems a critical point that can be briefly explained.
  5. Line 192, what does this mean? 5m above ground? [19] is not so clear about it either. Why 5m is chosen...Understory veg. ?
  6. Line 196, do you average the forest parameter values that correspond to the 50 best matched simulated waveforms? Any chance these 50 simulated waveforms give wide ranges of forest parameters and even disparate parameter values? That is, different forest conditions return the same lidar waveform.
  7. Line 200, after initinal quality flag filtering, you've got 110M GEDI waveforms to be matched. So every GEDI waveform found a match from the simulated waveforms? No failure of matching?
  8. Line 211, ODM is never spelled out…
  9. Section 3.1, why AGB differs much while GPP/NEE does not. I did not see this result being discussed in the Discussion.
  10. Line 233, “the estimated biomass”, you still mean aboveground biomass, not total of above+below, right? Be consistent here.
  11. 3 and Fig. 4, Is this RH95 value per each 1km grid based on all the best matched simulated waveforms or all the real GEDI waveforms in each 1km2 grid. How many GEDI waveforms per each 1km2 grid? Given a range or a distributiono of waveform counts over 1km2 grid? do you have to filter out grids that have too few GEDI waveforms?
  12. Line 282, you mean the GPP asymptotes at 150Mg/ha AGB? But sorry I can see this claim clearly from the Figure 5a. Please guide me through the figure to reach this observation.
  13. Line 285 – 294, The text here is really wordy, winding and difficult to read. Yet I think your figure 5a and 5b contain quite valuable information that is not stated clearly. If I'm interpreting the figure 5 correctly, the most pronounced informatiion is the link between age distribution and height heterogeneity (or complexity). At the same AGB, younger stands have higher GPP (5b) and stands with more homogeneous heights have higher GPP (5a). And vice versa. So lower structure complexity indicates a younger stands that explains higher GPP given the same AGB.
  14. Line 366, “these are probably”, what is "these"? stands with a heterogeneous height? If so, do you mean stands of higher species diversity have lower productivity? This claim seems counter intuitive to me. Can you find any reference to corrobarate your claim?

Round 2

Reviewer 2 Report

line 14, prob. "in great detail"

fig.7, normalized frequencies or absolute? this study, Roedig, MODIS all at 1km2 and same number of pixels?  unclear in the text.

line 400-402, I'm afraid that I wouldn't claim all three are in good agreement. Specify "good" here. What I can say is the distribution from this study is closer to the MODIS-based distribution than Roedig-based distrubtion.
